# A Comparative Machine Learning Study Identifies Light Gradient Boosting Machine (LightGBM) as the Optimal Model for Unveiling the Environmental Drivers of Yellowfin Tuna (*Thunnus albacares*) Distribution Using SHapley Additive exPlanations (SHAP) Analysis

**DOI:** 10.3390/biology14111567

**Published:** 2025-11-09

**Authors:** Ling Yang, Weifeng Zhou, Cong Zhang, Fenghua Tang

**Affiliations:** 1East China Sea Fisheries Research Institute, Chinese Academy of Fishery Sciences, Shanghai 200090, China; 2College of Information Engineering, Zhejiang Ocean University, Zhoushan 316022, China; 3Graduate School of Chinese Academy of Agricultural Sciences, Beijing 100081, China

**Keywords:** yellowfin tuna, catch per unit effort (CPUE), environmental drivers, machine learning, feature importance analysis, SHapley Additive exPlanations (SHAP), explainable machine learning, comparative analysis, Light Gradient Boosting Machine (LightGBM), ensemble learning

## Abstract

**Simple Summary:**

Tuna fisheries are a vital source of global protein, making it important to understand the key environmental factors that influence their distribution. This study aimed to identify which environmental conditions most affect where yellowfin tuna gather in the western and central Pacific Ocean. Using integrated fishing log data and 24 multi-source environmental variables, we applied and compared 16 machine learning regression models. The Light Gradient Boosting Machine (LightGBM) performed best and was selected to evaluate the influence of key environmental drivers. The results highlight that spatiotemporal and thermal factors are the most important predictors of tuna distribution. This research provides a reliable, data-driven framework to support sustainable fishery management, resource assessment, and operational forecasting.

**Abstract:**

Fishery resources of tuna serve as a vital source of global protein. This study investigates the key environmental drivers influencing the spatial distribution of yellowfin tuna (*Thunnus albacares*) in the western tropical Pacific Ocean. A comprehensive dataset was constructed by linking the catch per unit effort (CPUE) from 43 Chinese longline fishing vessels (2008–2019) with 24 multi-source environmental variables. To accurately model this complex relationship, a total of 16 machine learning regression models, including advanced ensemble methods like Light Gradient Boosting Machine (LightGBM), Random Forest, and Categorical Boosting Regressor (CatBoost), were evaluated and compared using multiple performance metrics (e.g., Coefficient of Determination [R^2^], Root Mean Squared Error [RMSE]). The results indicated that the Light Gradient Boosting Machine (LightGBM) model achieved superior performance, demonstrating excellent nonlinear fitting capabilities and generalization ability. For robust feature interpretation, the study employed both the model’s internal feature importance metrics and the SHapley Additive exPlanations (SHAP) method. Both approaches yielded highly consistent results, identifying temporal (month), spatial (longitude, latitude), and key seawater temperature indicators at intermediate depths (T450, T300, T150) as the most critical predictors. This highlights significant spatiotemporal heterogeneity in the distribution of *Thunnus albacares*. The analysis suggests that mid-layer ocean temperatures directly influence catch rates by governing the species’ vertical and horizontal movements. In contrast, large-scale climate indices such as the Oceanic Niño Index (ONI) exert indirect effects by modulating ocean thermal structures. This research confirms the dominance of spatiotemporal and thermal variables in predicting yellowfin tuna distribution and provides a reliable, data-driven framework for supporting sustainable fishery management, resource assessment, and operational forecasting.

## 1. Introduction

Fishery resources of tuna serve as a vital source of global protein and a key pillar supporting the economic development of coastal nations. Their sustainable exploitation and refined management have long been the focus of international attention [1]. Within the practices of resource assessment and fisheries management, fishery forecasting stands out as a core technological approach. Its primary objective is to dynamically predict the distribution of target species by modeling the response relationships between environmental factors and fishery resource abundance. This process involves not only multivariate and multi-scale ecological modeling but also demands that models possess robust spatiotemporal adaptability and explanatory power.

Yellowfin tuna is a highly migratory, large-bodied pelagic species of significant commercial value, widely distributed across tropical and subtropical oceans. It exhibits high sensitivity to environmental variability [2]. Previous studies have demonstrated that the habitat selection of *Thunnus albacares* is shaped by complex, nonlinear responses to a range of environmental variables, including sea surface temperature (SST), chlorophyll-a concentration (Chl-a), and large-scale climate indices [3]. However, traditional statistical models based on linear assumptions often fail to capture the synergistic interactions among multidimensional environmental variables and the inherent spatiotemporal heterogeneity of marine ecosystems [4]. As a result, these models exhibit limited predictive capacity and ecological interpretability, leading to substantial uncertainty in fishery forecasting outcomes.

Catch per Unit Effort (CPUE) is a key indicator for assessing fishery resource abundance. CPUE refers to the amount of fish caught per standardized unit of fishing effort. For tuna longline fisheries, the fishing effort is typically measured in terms of the number of hooks, usually in thousands of hooks. It not only reflects the dynamic changes in the target population but also indirectly captures the regulatory effects of environmental factors on fishing efficiency [5]. Although CPUE has been widely applied in fisheries science and resource assessment, current research on feature selection faces two primary challenges. First, modeling approaches remain largely dominated by traditional linear regression techniques, with insufficient systematic evaluation of emerging machine learning algorithms such as ensemble learning. In particular, there is a lack of comparative studies on their abilities to capture nonlinear responses and interactions among features [6]; Second, the selection of predictor variables often relies on expert-driven or empirical screening, with limited use of model-based variable importance metrics. This may result in the underestimation of key drivers or the overestimation of redundant variables, ultimately compromising the ecological interpretability and predictive accuracy of the results [7].

To address these issues, this study focuses on yellowfin tuna in the western and central Pacific Ocean and establishes a model comparison framework encompassing 16 regression algorithms to systematically evaluate the applicability and predictive performance of different approaches in CPUE modeling. The modeling suite includes a range of methodologies, covering linear regression, decision tree models, ensemble learning techniques, and multilayer perceptron (MLP) neural networks. Particular emphasis is placed on the Light Gradient Boosting Machine (LightGBM) algorithm, which effectively addresses common challenges in fishery datasets, such as sample imbalance, multicollinearity among variables, and spatiotemporal dependencies. In addition, SHapley Additive Explanations (SHAP) values and the built-in feature importance metrics of the models are used to analyze the nonlinear influence mechanisms and spatial heterogeneity of environmental factors affecting CPUE, thereby enhancing the ecological interpretability of model outcomes.

## 2. Materials and Methods

### 2.1. Data Processing

#### 2.1.1. Marine Environmental Data

This study focuses on the primary fishing grounds of longline fleets targeting *Thunnus albacares* in the western and central Pacific Ocean, specifically within the area bounded by 110° E to 170° W longitude and 0°to 30° S latitude. The fishery production data were obtained from the fishing logbooks of 43 distant-water longline vessels operated by the China National Fisheries Corporation (2008–2019). These logbooks contain key operational information, including vessel name, fishing date (year/month), fishing location (latitude and longitude), species composition, catch weight, number of individuals caught, and number of hooks deployed [8]. This dataset provides the foundational basis for constructing the CPUE indicator and analyzing its relationship with environmental variables. The spatial distribution of CPUE is shown in Figure 1.

The temporal dynamics of Catch Per Unit Effort (CPUE), plotted as monthly means with standard deviation ranges, revealed significant seasonal fluctuations (Figure 2). Specifically, CPUE values showed pronounced variability from May to August alongside rising temperatures, peaking in June and demonstrating a trend of higher CPUE during the warm spring and summer months.

The environmental variables used in this study were obtained from the following sources: Chlorophyll-a (Chl-a) data were retrieved from NASA’s Ocean Color remote sensing platform (https://oceancolor.gsfc.nasa.gov, accessed on 6 November 2025); Sea Level Anomaly (SLA) data were provided by AVISO (https://www.aviso.altimetry.fr, accessed on 6 November 2025); Eddy Kinetic Energy (EKE) and temperature–salinity profiles from 0 to 500 m were obtained from the Copernicus Marine Environment Monitoring Service (https://dataspace.copernicus.eu, accessed on 6 November 2025). For the climate indices, the Southern Oscillation Index (SOI) and Arctic Oscillation Index (AOI) were sourced from NOAA’s Climate Prediction Center (https://www.cpc.ncep.noaa.gov, accessed on 6 November 2025), the Pacific Decadal Oscillation Index (PDOI) was sourced from the National Centers for Environmental Information (NCEI) at NOAA (https://www.ncei.noaa.gov/access/monitoring/pdo/, the Pacific Decadal Oscillation Index (PDOI) was obtained from the the National Centers for Environmental Information (NCEI) at NOAA (https://www.ncei.noaa.gov/access/monitoring/pdo/, accessed on 6 November 2025), and the North Pacific Gyre Oscillation Index (NPGOI) was published via the Copernicus data platform (https://data.marine.copernicus.eu, accessed on 6 November 2025).

All environmental variables used in this study have a temporal resolution of one month. In terms of spatial resolution, sea level anomaly (SLA), eddy kinetic energy (EKE), and temperature–salinity profile data were provided at a 0.25° × 0.25° grid, while chlorophyll-a (Chl-a) data were available at a spatial resolution of 4 km. To ensure consistency in analytical scale, all environmental variables were resampled to a standardized grid of 0.5° × 0.5° using Python-based spatial processing tools (v3.10.2). These resampled datasets were then spatially matched and integrated with the fishing location data, enabling a joint analysis of catch variability and environmental drivers.

#### 2.1.2. Fishery Resource Abundance

Catch per unit effort (CPUE) is a key metric for evaluating fishing efficiency and resource abundance, and has been widely applied in fishery stock assessment and management studies [9]. To systematically analyze the spatiotemporal distribution of *Thunnus albacares* and the variability of its CPUE, the study area was divided into spatial grids of 0.5° × 0.5°. Based on longline logbook records, monthly statistics were compiled for each grid cell, including fishing effort (number of hooks deployed) and the number of individuals caught. These values were used to compute CPUE for each grid cell on a monthly basis. The CPUE was calculated using the following formula: where $CPUE_{\left(i,j\right)}$, $F_{fish\left(i,j\right)}$, $H_{hook\left(i,j\right)}$ represent, respectively, the monthly average CPUE (number of individuals per thousand hooks), the total number of fish caught, and the total number of hooks deployed in the grid cell located at the *i*-th longitude and the *j*-th latitude.(1)$$CPUE_{\left(i,j\right)}=\frac{F_{fish\left(i,j\right)}\times 1000}{H_{hook\left(i,j\right)}}$$

#### 2.1.3. Data Preprocessing

To analyze the relationship between catch per unit effort (CPUE) of *Thunnus albacares* and environmental variables, a comprehensive integration of multi-source datasets was first carried out. By spatially and temporally matching longline logbook records with environmental observation data, a unified dataset was constructed to ensure that each feature variable corresponded precisely to the observed CPUE at the same spatiotemporal coordinates. This process resulted in a curated dataset of 18,029 valid CPUE records. The CPUE values were then calculated, and relevant variables were screened accordingly. The final feature set consisted of 25 variables, encompassing three main categories: fishing operation parameters, oceanographic environmental factors, and climate anomaly indices. Specifically, these included:

(1) Fishing operation parameters: Catch per Unit Effort (CPUE) as a direct proxy for relative fish abundance, year (to account for long-term trends), month (to capture seasonal cycles), and latitude/longitude (to define the spatial context and static habitat features), formed the foundational data layer.

(2) Oceanographic environmental variables: Chlorophyll-a concentration (Chl-a), chlorophyll-a in the previous month (Chl_bf), chlorophyll-a in the following month (Chl_af); sea surface temperature in the previous month (SST_bf), and in the following month (SST_af); chlorophyll anomaly (Chldt), sea surface temperature anomaly (SSTdt), sea surface temperature gradient (SSTgrad), chlorophyll gradient (Chlgrad); sea level anomaly (SLA); eddy kinetic energy (EKE); and temperature at various depths including the surface (T0), 150 m (T150), 300 m (T300), and 450 m (T450).

Chlorophyll-a concentration (Chl-a) and its temporal lags (Chl_bf, Chl_af) served as indicators of primary production and the base of the food web. Sea surface temperature (SST_bf, SST_af) was included for its fundamental influence on physiological processes and thermal habitat suitability. Anomalies (Chldt, SSTdt) and horizontal gradients (Chlgrad, SSTgrad) of these two parameters were used to identify environmentally anomalous areas and productive frontal zones, which are known foraging hotspots. Furthermore, sea level anomaly (SLA) and eddy kinetic energy (EKE) were utilized as proxies for mesoscale ocean dynamics, such as eddies and currents, which affect prey aggregation and retention. Finally, temperature at various depths (T0, T150, T300, T450) characterized the vertical thermal structure, which is critical for defining the vertical habitat range and thermocline depth for pelagic species.

(3) Climate indices: Pacific Decadal Oscillation Index (PDOI), Southern Oscillation Index (SOI), Arctic Oscillation Index (AOI), North Pacific Gyre Oscillation Index (NPGIO), and the Oceanic Niño Index (ONI), which represents El Niño–Southern Oscillation (ENSO) phases. These indices modulate local oceanographic conditions, thereby exerting a bottom-up control on ecosystem productivity and species distributions over interannual to decadal timescales.

These features were categorized based on their environmental relevance to temperature, chlorophyll concentration, and oceanographic phenomena. As illustrated in Figure 3, the variables were grouped into six thematic categories: fishery indicators, ocean dynamics, thermal structure, temperature gradients, chlorophyll-related variables, and climate indices. Each subplot integrates kernel density estimation (KDE) curves with semi-transparent histograms, thereby simultaneously presenting the smoothed distribution trend and the frequency of the raw data.

The overlapping density curves allow for a visual comparison of intra-group parameter distributions, facilitating the identification of potential multicollinearity issues among variables. Meanwhile, the histogram frequency data provides insight into the spatiotemporal completeness of environmental sampling, particularly highlighting the prevalence of extreme values.

### 2.2. Research Methods

In this study, the catch per unit effort (CPUE) of *Thunnus albacares* was used as the response variable. Relevant fishery production records and environmental feature variables potentially influencing CPUE were collected and integrated into a comprehensive dataset. This dataset includes not only the observed CPUE values of *Thunnus albacares* but also a wide range of multidimensional environmental factors and climate indices that may affect fishing success. To ensure the robustness and validity of model training and evaluation, the complete dataset was randomly split into a training set (80%) and a testing set (20%) following an 8:2 ratio [10].

To comprehensively evaluate and identify the key factors influencing CPUE, this study employed a comparative framework involving 16 representative regression models. These included linear regression, ridge regression, Lasso regression, elastic net, random forest regression, extreme gradient boosting (XGBoost), and Light Gradient Boosting Machine (LightGBM), among others. Based on multi-model comparisons that integrated considerations of fitting accuracy, generalization capability, and feature interpretability, LightGBM was ultimately identified as the best-performing regression model.

LightGBM exhibits strong feature selection capabilities by automatically shrinking the coefficients of features that are irrelevant or weakly correlated with the response variable to zero. This effectively eliminates redundant variables and enhances overall model performance. Moreover, the algorithm demonstrates robustness in handling high-dimensional features, nonlinear relationships, and multicollinearity—making it particularly well-suited for modeling complex environmental datasets in this study.

To further optimize model performance, cross-validation was employed to tune hyperparameters such as regularization strength. The selection of the optimal model was based on the quantitative comparison of multiple error evaluation metrics, including mean squared error (MSE) and the coefficient of determination (R^2^). The workflow of the modeling process is illustrated in Figure 4.

#### 2.2.1. Introduction to Regression Algorithms

As a fundamental statistical tool, regression modeling plays a crucial role in uncovering relationships among variables and in predictive analytics. By establishing mathematical functions that link a dependent variable to one or more independent variables, regression models not only quantify the contribution of each factor to the response variable but also help to reveal potential causal mechanisms through parameter estimation techniques [11].

In this study, the marine environmental variables involved exhibit high dimensionality, heterogeneity, and complex nonlinear characteristics, making it challenging to comprehensively predict their effects on the catch per unit effort (CPUE) of *Thunnus albacares* using a single modeling approach. Therefore, to thoroughly assess the predictive performance of various regression models, a diverse set of representative supervised learning algorithms was employed. These models span several methodological categories, including linear models, instance-based learning methods, decision tree models, boosted tree models, ensemble learning techniques, and neural network-based approaches.

Linear models represent the most fundamental class of regression methods, based on the assumption of a linear relationship between input features and the target variable. These models are characterized by strong interpretability and computational efficiency. Among them, Linear Regression models the linear association between predictors and the response variable, featuring a simple structure and fast fitting and prediction speeds, making it suitable for linearly separable datasets [12].

Ridge Regression introduces an L2 regularization term to address multicollinearity issues, improving model stability and enhancing generalization performance by penalizing large coefficients [13]. Lasso Regression, incorporating L1 regularization in the loss function, effectively shrinks less relevant coefficients to zero, thereby performing automatic feature selection and providing resistance to overfitting [14].

The ElasticNet Regressor combines both L1 and L2 penalties, balancing feature selection and model robustness, and is particularly suitable for high-dimensional and sparse datasets [15]. Huber Regression, which employs the Huber loss function, provides robustness to outliers by reducing their influence while maintaining the linear structure of the model [16].

Neighbor-based learning methods construct predictions based on distance metrics among samples without assuming an explicit functional form, making them suitable for small-scale datasets. The K-Neighbors Regressor predicts target values by referencing the distance to neighboring training samples. This approach is simple and intuitive, requiring no prior assumptions about data distribution. However, its computational efficiency significantly decreases in large datasets or high-dimensional feature spaces [17].

Tree-based models operate by recursively partitioning the feature space to generate predictive rules, offering strong interpretability and the capability to model nonlinear relationships. Among them, the Decision Tree Regressor builds a hierarchical structure that splits the feature space into decision paths based on conditional rules. This method is easy to interpret and visualize, but is prone to overfitting, especially when dealing with noisy data [18].

The Extreme Gradient Boosting Regressor (XGBoost Regressor) improves upon traditional boosting techniques by incorporating regularization and parallel optimization strategies within a gradient boosting framework. These enhancements significantly increase both model accuracy and computational efficiency, making XGBoost widely adopted in structured data modeling tasks [19].

Boosted Tree Models utilize the boosting mechanism, which enhances overall predictive performance by iteratively combining multiple weak learners. The Light Gradient Boosting Machine Regressor (LightGBM Regressor) employs a histogram-based optimization strategy within the gradient boosting framework, substantially reducing memory usage and training time. This makes it particularly suitable for large-scale datasets [20]. The Categorical Boosting Regressor (CatBoost Regressor) is specifically optimized for handling categorical features by automatically performing encoding transformations, effectively mitigating overfitting issues. It is well-suited for modeling tasks involving a large number of categorical variables [21].

Ensemble Learning Methods improve overall model performance by aggregating the predictions of multiple base learners, thereby effectively reducing both variance and bias. The Random Forest Regressor builds an ensemble of decision trees using randomly sampled training subsets and aggregates their outputs via averaging or voting. It is known for its strong robustness and resistance to overfitting [22]. The Adaptive Boosting Regressor (AdaBoost Regressor) iteratively trains weak learners by reweighting samples, making it suitable for capturing complex nonlinear relationships, although it tends to be sensitive to outliers [23]. The Gradient Boosting Regressor optimizes the model through residual learning, achieving high predictive accuracy; however, it requires careful hyperparameter tuning and incurs relatively high computational cost during training [24].

Extreme Ensemble Methods enhance model diversity and generalization by introducing greater randomness on top of standard ensemble strategies. The Extremely Randomized Trees Regressor (ExtraTrees Regressor) increases diversity and robustness by randomly selecting both features and split points during tree construction [25]. The Bagging Regressor trains multiple base learners independently on different bootstrapped subsets of the training data and aggregates their predictions, effectively reducing variance and improving model stability [26].

Neural Network Models possess strong nonlinear modeling capabilities, making them suitable for handling complex or high-dimensional data structures. The Multilayer Perceptron Regressor (MLP Regressor) constructs a deep feedforward neural network and uses nonlinear activation functions to capture intricate relationships between inputs and outputs. However, its performance can be highly sensitive to training data quality and hyperparameter settings [27].

#### 2.2.2. Model Performance Rating

(1) The Pearson correlation coefficient is a statistical measure used to quantify the strength and direction of the linear relationship between two continuous variables. Its value ranges from −1 to 1. As one of the most commonly used correlation coefficients, it is widely applied in scientific research, data analysis, social sciences, and engineering domains [28]. The calculation formula is given below, where *N* denotes the number of data points; *x_i_* and *y_i_* represent the *i*-th observations of variables *x* and *y*, respectively; *i* is the sample index; $\overset{-}{x}$ and $\overset{-}{y}$ denote their corresponding sample means:(2)$$\rho_{XY}=\frac{\sum_{i=1}^{N}\left(x_{i}-\bar{x}\right)\left(y_{i}-\bar{y}\right)}{\sqrt{\sum_{i=1}^{N}{\left(x_{i}-\bar{x}\right)}^{2}}\cdot \sqrt{\sum_{i=1}^{N}{\left(y_{i}-\bar{y}\right)}^{2}}}$$

(2) The Jensen–Shannon Divergence (JSD) is a symmetric metric used to quantify the similarity between two probability distributions, *P* and *Q*. Unlike the Kullback–Leibler divergence, JSD is bounded and more stable, with values ranging from 0 to 1 [29]. The formula is defined as follows, where *P* and *Q* represent the two probability distributions, and *H*(*P*) denotes the entropy of *P*:(3)$$JSD\left(P∥Q\right)=H\left(\frac{P+Q}{2}\right)-\frac{1}{2}\left[H\left(P\right)+H\left(Q\right)\right]$$

(3) The Mean Absolute Error (MAE) is used to quantify the average magnitude of deviations between predicted values and actual observations, offering an intuitive interpretation of prediction accuracy [30]. The formula for MAE is defined as follows, where *n* represents the number of samples, *i* is the sample index, *f_i_* denotes the predicted value, and *y_i_* is the corresponding observed value:(4)$$MAE=\frac{1}{n}\sum_{i=1}^{n}\left|f_{i}-y_{i}\right|$$

(4) The Mean Squared Error (MSE) measures the average of the squared differences between predicted values and actual observations. It penalizes larger errors more heavily, making it more sensitive to significant deviations in prediction [31]. A lower MSE indicates that the predicted values are closer to the actual values, implying better model fit. The formula for MSE is given as follows: where *n* represents the number of samples, *i* is the sample index, *f_i_* denotes the predicted value, and *y_i_* is the corresponding observed value:(5)$$MSE=\frac{1}{n}\sum_{i=1}^{n}{\left(f_{i}-y_{i}\right)}^{2}$$

(5) The Root Mean Squared Error (RMSE) extends the concept of MSE by taking the square root of the average squared errors. It is particularly sensitive to large deviations, making it effective at highlighting the impact of outliers in prediction performance. The formula for RMSE is as follows, where RMSE is the square root of the MSE:(6)$$RMSE=\sqrt{\frac{1}{n}\sum_{i=1}^{n}{\left(f_{i}-y_{i}\right)}^{2}}$$

(6) The Explained Variance Score (EVS) evaluates the proportion of the variance in the observed data that is captured by the predictive model, indicating the model’s effectiveness in explaining data variability [32]. The formula for EVS is as follows, where *y* denotes the observed values, f the corresponding predicted values, and Var represents the variance of *y*:(7)$$EVS=1-\frac{Var\left(y-f\right)}{Var\left(y\right)}$$

(7) The Coefficient of Determination (R^2^) is another key goodness-of-fit metric that quantifies the proportion of variance in the dependent variable that is explained by the regression model. It is commonly used to assess the overall performance of predictive [33]. In the context of regression analysis, it reflects the degree to which the independent variables account for the variability in the dependent variable. The formula for R^2^ is as follows: where *n* is the number of samples, *i* is the sample index, *y_i_* is the actual observed value, *f_i_* is the predicted value, and $\overset{-}{y}$ is the mean of the actual observed values.(8)$$R^{2}=1-\frac{\sum_{i=1}^{n}{\left(y_{i}-f_{i}\right)}^{2}}{\sum_{i=1}^{n}{\left(y_{i}-\bar{y}\right)}^{2}}$$

(8) To comprehensively evaluate the performance of the regression models, this study employed five key evaluation metrics: MAE, EVS, MSE, RMSE, and R^2^, and further derived a Composite Score. First, all metric values were normalized to ensure consistency in scale. For MAE, MSE, and RMSE, where lower values indicate better model performance, reverse normalization was applied. For EVS and R^2^, where higher values indicate better performance, direct normalization was used. Next, weighted aggregation was performed based on the relative importance of each metric. MAE, MSE, and RMSE were each assigned a weight of 0.5/3, while EVS and R^2^ were each assigned a weight of 0.25, summing to a total weight of 1 [10]. The Composite Score was calculated using the following formula, where $X_{i}$ represents the normalized score of the *i*-th model for a given metric.(9)$$\mathrm{Score}=\left(\sum_{i\in \{MAE,\;MSE,\;RMSE\}}\left(1-\frac{x_{i}-\min\left(x_{i}\right)}{\max\left(x_{i}\right)-\min\left(x_{i}\right)}\right)\times \frac{0.5}{3}\right)+\left(\sum_{i\in \{EVS,\;R^{2}\}}\frac{x_{i}-\min\left(x_{i}\right)}{\max\left(x_{i}\right)-\min\left(x_{i}\right)}\times 0.25\right)$$

#### 2.2.3. Variable Screening and Feature Selection

In regression modeling, feature selection is a critical step for enhancing model performance, improving interpretability, reducing computational complexity, and preventing overfitting. This process not only influences the predictive accuracy of the final model but also sheds light on the relative importance of environmental drivers influencing the abundance of *Thunnus albacares*. In this study, feature selection was conducted using machine learning-based regression models, integrating Catch Per Unit Effort (CPUE) data of *Thunnus albacares* with 25 environmental and climatic variables. The feature selection procedures were implemented in Python (v3.10.2), utilizing packages such as Pandas (v2.2.2), NumPy (v1.24.4), Matplotlib (v3.5.3), and the machine learning library scikit-learn (v1.1.2). The overall workflow was structured into the following four stages:

(1) Data Preprocessing Stage: In the initial phase, 25 environmental features potentially associated with CPUE variations in *Thunnus albacares* were extracted from multiple data sources. These variables included spatial coordinates (latitude and longitude), temporal variables (year and month), remote sensing variables (e.g., SST, Chl-a, SLA, EKE), and major climatic indices (e.g., PDOI, SOI, AOI, ONI). Data were imported, integrated, and preliminarily cleaned using the Pandas library in Python. Subsequently, the 24 environmental variables were designated as independent variables, and the CPUE of *Thunnus albacares* was defined as the dependent variable. The full dataset was randomly divided into training (80%) and testing (20%) subsets to support the modeling and feature importance analysis.

(2) Model Parameter Optimization Stage: Model performance in regression tasks is highly dependent on appropriate hyperparameter settings. To improve predictive capability, the Light Gradient Boosting Machine (LightGBM) was selected as the core modeling algorithm, and parameter tuning was conducted using a randomized search strategy (RandomizedSearchCV). The optimization covered 15 key parameters involving aspects of tree structure (e.g., maximum depth and number of leaves, which control model complexity), regularization strength (e.g., L1 and L2 penalties to prevent overfitting), and training control (e.g., learning rate and number of estimators to balance training speed and accuracy). A five-fold cross-validation (5-fold CV) approach was used to evaluate the generalization performance of each parameter combination, and the one yielding the lowest average error was selected as the optimal setting.

(3) Feature Importance Analysis Stage: Upon completing model training, the intrinsic feature importance rankings generated by the LightGBM model were extracted. These rankings, derived from the model’s internal decision tree structure, indicate the relative contribution of each feature to the overall prediction accuracy. By analyzing the feature importance scores, key predictors with high explanatory power were identified, while weakly correlated or redundant variables were excluded through dimensionality reduction. This process not only improved model efficiency and generalization capacity, but also enhanced interpretability, offering valuable insights for subsequent ecological mechanism analysis.

(4) SHAP (SHapley Additive exPlanations) Analysis Stage: To further enhance model interpretability and transparency, this study incorporated SHAP, a game-theoretic feature attribution method. Originating from Shapley values in cooperative game theory, SHAP aims to calculate the marginal contribution of each feature to the model output across all possible feature combinations, averaging these contributions to yield an accurate and fair representation of each variable’s influence [34]. It has been widely applied in ecological model interpretation, environmental risk assessment, and policy response analysis. Unlike LightGBM’s feature importance ranking, based on split frequency and split gain within the tree structure and reflecting structural feature contributions, SHAP values quantify each feature’s direct contribution to individual predictions. This facilitates the generation of more informative and interpretable feature importance rankings, offering insights into how each environmental factor influences the predicted CPUE values for *Thunnus albacares* [35].

## 3. Environmental Drivers Identification

This section presents the results of the model evaluation and selection process. The performance of 16 machine learning models is systematically compared through visual assessments and quantitative metrics to identify the optimal model for predicting *Thunnus albacares* CPUE. Subsequently, feature selection and an explanation of the final model’s predictions are provided.

### 3.1. Comparison Between Actual and Predicted Values

When performing predictive tasks, selecting an appropriate machine learning model is crucial. To this end, the study compared the prediction results of 16 models using kernel density estimation (KDE) plots, as illustrated in Figure 5. The figure presents the performance of various machine learning models, where each subplot depicts the relationship between the predicted and observed values. Color gradients, ranging from blue (low density) to red (high density), represent the density distribution of data points and provide an intuitive understanding of how each model fits the data across different regions.

The diagonal line in each subplot indicates the ideal condition where predicted values equal observed values, serving as a visual benchmark for assessing model performance. This comparative visualization provides both qualitative and quantitative insights into the models’ fitting capabilities, supporting the identification of the most suitable algorithm for subsequent analyses related to *Thunnus albacares* catch per unit effort (CPUE) prediction.

The study evaluated model performance using three metrics: the coefficient of determination (R^2^), Pearson correlation coefficient, and Jensen–Shannon divergence (JSD). Among them, R^2^ and Pearson’s coefficient are the most commonly used indicators in regression analysis, which assess model goodness-of-fit and the linear correlation between predicted and observed values, respectively. In contrast, JSD measures the similarity between the distributions of predicted and actual values; a smaller JSD indicates a closer match between the two distributions, reflecting better agreement at the population level.

As shown in Figure 5, some models exhibit significant deviations from the diagonal line, indicating systematic errors in specific value ranges. In comparison, ensemble learning methods such as LightGBM and ExtraTrees demonstrate superior performance in terms of prediction concentration and alignment along the diagonal. The high density of scatter points near the diagonal line suggests strong predictive accuracy and consistency in distribution, which is essential for reliable modeling of *Thunnus albacares* catch per unit effort (CPUE).

### 3.2. Comparison of Regression Models’ Performance

To further visualize and compare the performance of different models, this study employed a radar chart based on the standardized R^2^ scores, enabling an intuitive comparison of various machine learning algorithms and providing data-driven support for subsequent model selection and optimization. As shown in Figure 6, the values along the axes represent the performance scores of each model. The color gradient indicates the magnitude of the score, with higher scores represented by a deeper purple hue. The radar chart clearly illustrates the performance distribution across different evaluation metrics for each algorithm, highlighting that LightGBM demonstrates the most outstanding performance among all evaluated models.

To comprehensively evaluate model performance, we applied a multi-dimensional visualization strategy, as illustrated in Figure 7. Panel (a) shows R^2^ values along with 95% confidence intervals obtained through bootstrap resampling (*n* = 200), demonstrating considerable performance variation across the 16 machine learning algorithms examined. The horizontal layout enables direct comparison with the baseline model, indicated by a red dashed line. Panel (b) presents corresponding RMSE values and associated confidence intervals, offering complementary insight into predictive accuracy. In Panel (c), the improvement in R^2^ relative to the top-performing baseline model is quantified, with color coding used to distinguish positive (green) and negative (red) performance differences. Finally, Panel (d) summarizes the overall performance distribution via a graded classification system based on R^2^ improvement (Excellent: >0.1, Good: >0.05, Fair: >0.01, Poor: ≤0.01), providing an intuitive assessment of methodological effectiveness.

### 3.3. Comprehensive Scores

The evaluation of model performance typically relies on multiple metrics. To comprehensively compare the advantages and disadvantages of different models, this study employed five key evaluation indicators: Mean Absolute Error (MAE), Mean Squared Error (MSE), Root Mean Squared Error (RMSE), Explained Variance Score (EVS), and the Coefficient of Determination (R^2^). A detailed comparison of the scores across all 16 regression models and their performance under different evaluation criteria is presented in Table 1.

To facilitate a more intuitive and integrated assessment of overall performance, a composite score (Score) was introduced. This score was calculated by applying weighted aggregation to the five-evaluation metrics. The model with the highest composite score was selected as the optimal model and subsequently used for the feature selection process. The results, as illustrated in Figure 8, present a ranked summary of the overall scores for each regression model.

The findings indicate that the Light Gradient Boosting Machine (LightGBM) outperformed all other models, achieving the highest composite score. LightGBM demonstrated a superior capacity to handle large-scale datasets and effectively manage feature learning processes, particularly when dealing with complex data structures. By employing a composite scoring approach, this study not only revealed the strengths and weaknesses of each algorithm based on individual evaluation metrics but also highlighted their overall performance under multidimensional assessment criteria. Consequently, LightGBM was selected as the optimal model for subsequent feature selection and analysis.

### 3.4. Feature Selection of LightGBM

The strength and direction of linear relationships between numerical variables were assessed using Pearson correlation coefficients. These coefficients range from −1 to +1, representing a spectrum from perfect negative to perfect positive linear correlation. A correlation matrix was constructed and visualized as a heatmap via the Seaborn (v0.13.2) heatmap function. The matrix is displayed as a color-coded grid where the color intensity corresponds to the correlation strength, following a divergent color scheme (blue for negative and red for positive values) for clarity. The results of the correlation analysis between CPUE and environmental factors are shown in Figure 9.

In machine learning regression models, the objective of feature selection using LightGBM is to achieve triple optimization by enhancing predictive accuracy, improving model interpretability, and reducing computational complexity through the elimination of redundant features and irrelevant variables. This study employs a hierarchical control strategy to accomplish this goal. First, the growth of decision trees is constrained by setting thresholds for the number of leaf nodes and maximum depth, which helps suppress overfitting while increasing the model’s sensitivity to key features. Second, the use of random features and sample subset sampling introduces probabilistic filtering to mitigate noise interference in the training data, thereby enhancing the model’s generalization capability. Finally, the inclusion of regularization constraints effectively suppresses the weights of irrelevant features, improving model robustness and its capacity to manage complex feature interactions.

The prediction results demonstrate that the use of a “low learning rate with deep ensemble” strategy (learning rate = 0.01, number of trees = 800) effectively balances training stability and generalization capacity. The dual regularization framework (L1 = 0.1, L2 = 10) successfully suppresses the risk of overfitting in complex marine environments. In feature engineering, 90% dynamic feature sampling combined with gain-based evaluation enhances model robustness while preserving the representational power of the variables. The chosen tree structure parameters (number of bins = 700, number of leaves = 35) optimize computational accuracy and efficiency. Strict control of randomness (random seed = 42) ensures model reproducibility and verifiability in aquaculture-related environmental data analysis. As a result of comprehensive optimization, the model’s validation performance improved to 0.255. The detailed configuration of optimal LightGBM parameters is presented in Table 2.

LightGBM provides a direct and efficient approach for feature selection, making it well-suited for applications requiring rapid processing of large-scale datasets and accurate prediction. The results of feature selection using the optimal LightGBM parameters are presented in Figure 10, where the importance scores reflect the contribution of each key variable to the prediction of Catch Per Unit Effort (CPUE) for *Thunnus albacares*.

As shown in the bar chart, the importance of features varies significantly, with the variable “month” exhibiting the highest importance, far exceeding other features. To quantitatively assess the model’s predictive performance on CPUE under varying numbers of selected features, and thereby achieve a scientific and reasonable balance between feature reduction and model simplification, a line graph displays the R^2^ value and its 95% confidence interval. The R^2^ increases rapidly as more features are added, indicating that the inclusion of new variables significantly enhances model performance. However, the performance gain plateaus after the first 13 features. A vertical dashed line in the figure marks the top 13 features, indicating that this subset was selected as the most informative group based on both feature importance and model accuracy.

The final ranking of feature importance identified by LightGBM is as follows: month, lat, T450, T150, T300, NPGIO, year, SLA, lon, PDOI, SSTgrad, ONI, SST_bf, Chldt, AOI, SSTdt, Chl_bf, Chl-a, SST_af, Chl_af, EKE, T0, SOI, and Chlgrad.

LightGBM provides a direct and efficient approach for feature selection, making it well-suited for applications requiring rapid processing of large-scale datasets and accurate prediction. The results of feature selection using the optimal LightGBM parameters are presented in Figure 6, where the importance scores reflect the contribution of each key variable to the prediction of Catch Per Unit Effort (CPUE) for *Thunnus albacares*.

Model validation is essential for assessing the robustness of machine learning approaches like LightGBM. We systematically evaluate the key statistical assumptions using diagnostic plots in Figure 11. The Q-Q plot (a) and residual distribution (b) reveal severe non-normality, characterized by heavy tails (Shapiro–Wilk W = 0.658, *p* < 0.001), high skewness (4.92), and kurtosis (52.09). In contrast, the ACF plot (c) and Ljung–Box test (*p* = 0.307) confirm error independence, a finding supported by the random, unsystematic pattern in the residual sequence (d). Despite a well-calibrated mean residual near zero (−0.0101), the presence of extreme values (e) suggests potential outliers. These results validate the model’s error independence but highlight a significant deviation from normality, guiding future refinements.

Through integrated visualization techniques including Q-Q plots, residual distribution analysis, autocorrelation function plots, and residual sequence examination, we provide a comprehensive assessment of normality, independence, homoscedasticity, and autocorrelation assumptions. The findings demonstrate both strengths and limitations in the current modeling approach, offering valuable insights for methodological refinement.

### 3.5. SHAP Analysis

To elucidate sample-level decision mechanics of the LightGBM model, Figure 12 employs SHAP (SHapley Additive exPlanations) decomposition to visualize prediction pathways for *Thunnus albacares* catch per unit effort (CPUE). The horizontal axis traces cumulative changes in predicted CPUE values from the baseline expectation to final outputs, while the vertical axis ranks features by their global contribution magnitude.

Each line in the plot corresponds to a single sample and starts from the expected value (i.e., the mean predicted value across the training dataset). Along the path, SHAP values for each feature are added sequentially, ultimately reaching the model’s final prediction output. The horizontal axis shows the cumulative contribution of SHAP values from left to right; the vertical axis displays feature names, automatically ordered by their overall contribution to the prediction. Each bend or inflection point in a line indicates the extent to which a specific feature has a positive or negative influence on the predicted CPUE for that particular sample. The color intensity of each line reflects the density of overlapping samples, allowing for the visual identification of local patterns.

This visualization demonstrates the model’s step-by-step inference process for individual predictions. It can be used to identify consistent decision-making patterns and to uncover the underlying causes of prediction anomalies, serving as a critical link between global feature importance and local interpretability in ecological modeling.

To further analyze the SHAP-based feature importance ranking, this study employed a dual-axis visualization approach that combines the SHAP beeswarm plot with a bar chart of feature importance, allowing for an intuitive representation of each feature’s contribution and impact on the model’s prediction of *Thunnus albacares* CPUE.

As shown in Figure 13, the variable month exhibited the highest explanatory power, with most of its SHAP values being positive. This indicates that the month generally has a positive influence on the predicted CPUE. The variables latitude (lat) and longitude (lon) followed in importance, also demonstrating a strong contribution to model predictions. Several oceanographic and climate-related variables also showed notable explanatory capability. In contrast, variables such as year, NPGIO, and T150 displayed more dispersed SHAP value distributions, with wide variation in the direction and magnitude of their effects across different samples. This suggests a degree of uncertainty or inconsistency in their influence on CPUE. Variables such as EKE, T0, and chlgrad exhibited low average SHAP values, indicating weak influence on the model’s predictions, with some contributing negligibly across the dataset.

## 4. Discussion

### 4.1. Comparative Analysis of Different Regression Models

A comparative analysis of the 16 regression models revealed that the boosted tree model LightGBM and the extreme ensemble method ExtraTrees demonstrated excellent performance across all evaluation metrics. In particular, both models significantly outperformed others in terms of Mean Squared Error (MSE) and Mean Absolute Error (MAE), while also ranking among the top in Explained Variance Score (EVS) and the Coefficient of Determination (R^2^), showing superior overall predictive capabilities.

LightGBM, through its gradient-based histogram optimization algorithm, leaf-wise growth strategy, and native handling of categorical variables, significantly reduces the time complexity of model training and effectively mitigates overfitting, making it especially suitable for data scenarios involving large sample sizes, high-dimensional features, and strong multicollinearity [36].

In comparison, ExtraTrees improves model diversity and generalization by introducing an extremely randomized splitting strategy. It demonstrates greater robustness against common issues in fishery observational datasets, such as measurement errors, outliers, and spatial sampling imbalances [37]. Both models achieve a favorable balance between accuracy and stability, highlighting their high practical value in modeling the catch per unit effort (CPUE) of *Thunnus albacares*.

In addition, CatBoost maintains stable performance even under significant data heterogeneity or complex variable interactions by incorporating target encoding for categorical variables and an ordered boosting strategy. This makes it particularly suitable for fishery datasets where categorical variables are frequently coupled with temporal information [38].

Random Forest improves model robustness and resistance to noise by integrating multiple low-correlation decision trees. It maintains good predictive accuracy even under conditions of feature redundancy or suboptimal data quality [39], making it well-suited for ecology-oriented modeling tasks and the generation of policy recommendations where model interpretability is a priority [40].

Finally, in this study, models such as Bagging, Gradient Boosting, K Nearest Neighbors, and XGBoost demonstrated moderate performance, suggesting their potential applicability under specific conditions. In contrast, traditional linear models, including Lasso, Ridge, ElasticNet, and standard Linear Regression, exhibited relatively low overall performance. These models showed evident underfitting when faced with nonlinear ecological response mechanisms commonly observed in fishery systems. Their modeling capacity remains constrained by the linear framework, indicating structural limitations in capturing the complex relationships between oceanographic environmental factors and catch per unit effort (CPUE) of species such as *Thunnus albacares* [41]. Models ranked at the lower end of performance included MLP Regressor, Huber, AdaBoost, and Decision Tree. The MLP Regressor’s performance was likely constrained by the limited dataset size, which may be insufficient for optimizing its numerous parameters and complex architecture, leading to suboptimal convergence. The Huber regressor, while robust to outliers, may lack the flexibility to capture the full spectrum of nonlinear relationships present in the ecological data. Both AdaBoost and the standalone Decision Tree are particularly prone to overfitting and sensitivity to noise in the dataset; AdaBoost can amplify errors from weak learners, while the Decision Tree model easily learns spurious patterns in the training data, resulting in poor generalization to unseen data.

### 4.2. Comparative Analysis of Feature Selection

Analysis of the CPUE prediction model for *Thunnus albacares* identified a set of key spatiotemporal and environmental drivers. The results show a strong consistency between the feature ranking derived from LightGBM and the SHAP-based importance order. Temporal (month) and spatial features (latitude and longitude) exerted the greatest influence on the model’s predictive accuracy, followed by water column temperature features (T450, T300) and large-scale indices such as ONI and PDOI. In addition, variables such as year and NPGIO exhibited both positive and negative effects on the model predictions depending on context, while features like EKE, T0, and chlgrad had relatively minor impacts. Among all predictors, temperature-related environmental factors accounted for the largest proportion and consistently ranked among the top in terms of importance during the feature selection process.

As shown in the results, the month regulates sea temperature dynamics while latitude determines thermal gradients, creating combined effects on water mass distribution that exceed their individual impacts. Furthermore, large-scale climate indices (ONI, PDOI) interact with local thermal conditions in ways that substantially modify fish habitat suitability. For instance, the effect of mid-layer temperatures (T150, T300, T450) on vertical fish distribution is modulated by these climate oscillations, which alter thermal stratification patterns. These interaction mechanisms primarily function through their collective impact on ocean temperature—climate indices shape large-scale thermal regimes while spatial and temporal factors determine how these thermal patterns translate into local habitat conditions, explaining why variables like month and latitude consistently rank high in feature importance as they represent the spatiotemporal integration of these complex interactions.

In the multivariable interaction analysis, month and latitude consistently ranked among the top features in terms of importance. The variable month indirectly influences CPUE by regulating sea temperature and the spatial dynamics of water masses. This finding aligns with the study by Lan et al., which identified seasonal variation and latitudinal thermal gradients as key factors affecting fluctuations in *Thunnus albacares* catch rates [42], suggesting that seasonal changes significantly impact ocean temperature variability. Moreover, this finding aligns with the pattern demonstrated in Figure 2, where higher CPUE values coincide with the elevated temperatures of spring and summer months. Matsubara et al. reported that thermal differences across latitudes directly affect the geographic distribution of *Thunnus albacares* [43]. The coupling between latitude and large-scale oceanic thermal structures reflects a stable association with fishing ground spatial patterns [44]. These results are consistent with the known behavioral response of *Thunnus albacares* to thermal gradients, supporting the ecological hypothesis that this species exhibits pronounced seasonal migratory behavior and adapts to tropical and subtropical water masses [45]. Therefore, the high importance of these features underscores the central role of temperature-related factors in the prediction of marine fishery resources [10], consistent with numerous previous studies.

The study also revealed that mid to deep-layer temperature variables, such as T150, T300, and T450, exhibited significantly higher importance to model performance compared to surface-layer variables. These factors directly influence the vertical distribution and migratory pathways of fish species [46], and *Thunnus albacares* is known to exhibit a specific preference for certain depths within the vertical water column [47]. This depth preference likely reflects its adaptive response to optimal temperature layers, thermocline structures, and the distribution of midwater prey resources. These findings are consistent with the observational research by Song et al. on the thermocline-associated behavioral patterns of *Thunnus albacares* [48].

Oceanic anomaly indicators such as the North Pacific Gyre Oscillation Index (NPGIO), Oceanic Niño Index (ONI), and Pacific Decadal Oscillation Index (PDOI) are, to some extent, influenced by temperature fluctuations. In terms of climatic drivers, ONI, PDOI, and NPGIO exhibited relatively high average SHAP values, indicating a strong influence on the model output. This suggests that catch per unit effort (CPUE) is substantially regulated by large-scale climate systems, which may indirectly affect the spatial distribution and catchability of *Thunnus albacares* by altering thermal stratification, primary productivity, and current patterns [49]. Therefore, incorporating multi-scale climatic indices into CPUE modeling is essential, a conclusion supported by several regional fishery studies [3,50].

In contrast, some variables, such as the sea surface chlorophyll gradient (chlgrad), eddy kinetic energy (eke), and sea surface temperature anomalies (sst_af) ranked lower in the feature importance analysis. Variables including sst_bf, sstdt, sst_af, and T0 reflect horizontal variations in sea surface temperature, which are known to affect phytoplankton growth and, through their influence on water mass movement, indirectly alter nutrient availability and fish foraging behavior [51]. *Thunnus albacares* is known to exhibit specific ecological temperature thresholds. Given the relatively high variability and low stability of surface-layer temperatures, their influence on CPUE is limited. In contrast, temperature variations provide more stable signals that align better with the habitat preferences of *Thunnus albacares*, thereby demonstrating stronger explanatory power for CPUE, consistent with the findings of Wright et al. [52].

In summary, the prediction of *Thunnus albacares* catch per unit effort (CPUE) is not solely dependent on individual ecological factors, but rather results from the coupling of multi-scale temporal and spatial drivers. Temporal variables, latitude, and mid-layer ocean temperature emerge as high-frequency driving forces, while large-scale climatic oscillations provide low-frequency background disturbances. These findings further support the application value of machine learning models that integrate nonlinear predictive capacity with interpretability mechanisms in fishery resource assessments. Such approaches enhance the precision of ecological forecasting and contribute to more scientifically grounded policy formulation.

## 5. Conclusions

This study developed an analytical framework integrating multiple machine learning regression algorithms to systematically evaluate the performance of sixteen mainstream models in predicting the catch per unit effort (CPUE) of *Thunnus albacares*. The results demonstrated that the Light Gradient Boosting Machine (LightGBM) and Extremely Randomized Trees (ExtraTrees) models achieved superior performance across all regression metrics, exhibiting excellent capabilities in nonlinear fitting and capturing complex interactions among ecological drivers, thus showing strong application potential. CatBoost and Random Forest also displayed robust performance and high interpretability, making them well-suited for fisheries prediction scenarios involving heterogeneous ecological variables or requiring clear ecological inference. Based on feature importance rankings and SHapley Additive exPlanations (SHAP) analysis, the study further identified month, latitude, and multi-depth temperature variables as key factors driving the spatiotemporal variability of CPUE, highlighting the high sensitivity of *Thunnus albacares* to thermal and seasonal environmental fluctuations. These findings contribute to a deeper understanding of the relationships between environmental forcing mechanisms and the spatial dynamics of fishery resources.

While this study provides a robust modeling framework for understanding the environmental drivers of yellowfin tuna CPUE for the Chinese distant-water longline fishery, several limitations should be considered when interpreting the results. The primary limitation stems from our reliance on a single fleet source (Chinese longliners). Although this ensures internal consistency by controlling for vessel type, gear, and broad operational strategies, it may limit the immediate generalizability of our specific model predictions to other fleets (e.g., purse seiners, gillnetters) or different maritime regions. It is plausible that our model has learned relationships influenced by the particular operational preferences of this fleet, which might not fully transfer to other systems with distinct fishing tactics and target species compositions. Furthermore, the non-stationary nature of marine ecosystems under a changing climate also implies that the identified relationships between environmental drivers and CPUE may evolve over time.

These limitations define clear pathways for future research. First and foremost, a critical next step is to conduct a comparative analysis by applying the same modeling framework to datasets from diverse fleets and regions. Investigating how the importance and functional response of environmental drivers (e.g., sea surface temperature, chlorophyll-a) vary across different vessel types (e.g., longliners vs. purse seiners) and operational protocols would significantly enhance our understanding of the universality or context-dependence of these relationships.

Secondly, integrating higher-resolution oceanographic physical and chemical environmental data, along with individual behavioral datasets, could enhance the model’s ability to disentangle complex multi-factor coupling mechanisms and deepen the identification of ecological drivers. In addition, emerging modeling approaches such as transfer learning and federated learning hold significant potential in data-sparse regions, offering improved model robustness under extreme climate event conditions and enhancing the capacity to capture ecological responses at global scales. These advancements are expected to provide more scientific and reliable technical support for sustainable fisheries management and resource assessment related to Thunnus albacares and other marine species.

## Figures and Tables

**Figure 1 biology-14-01567-f001:**
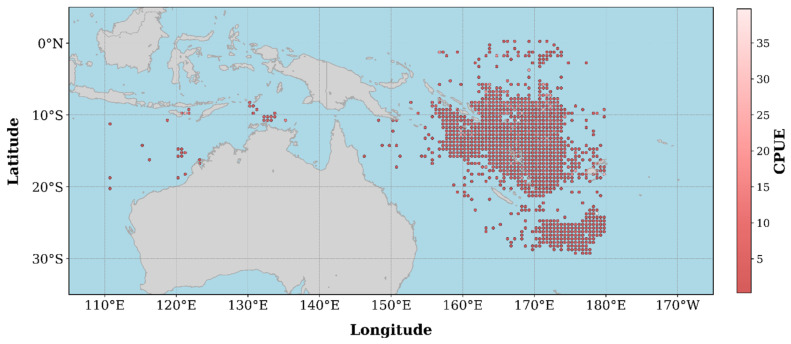
CPUE distribution in the western and central Pacific Ocean.

**Figure 2 biology-14-01567-f002:**
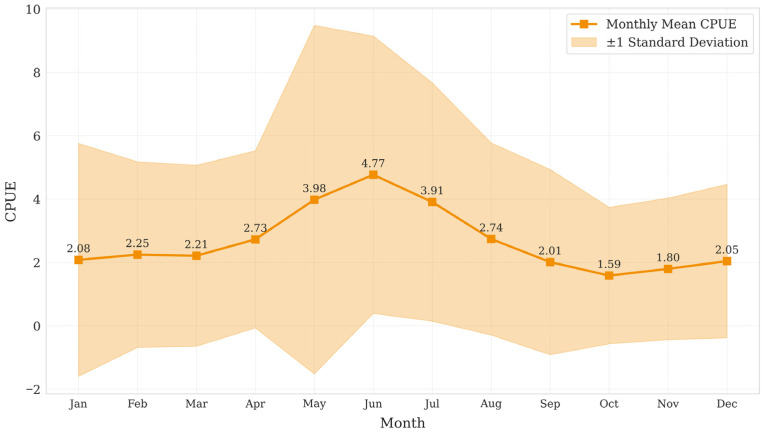
CPUE monthly means data change line chart.

**Figure 3 biology-14-01567-f003:**
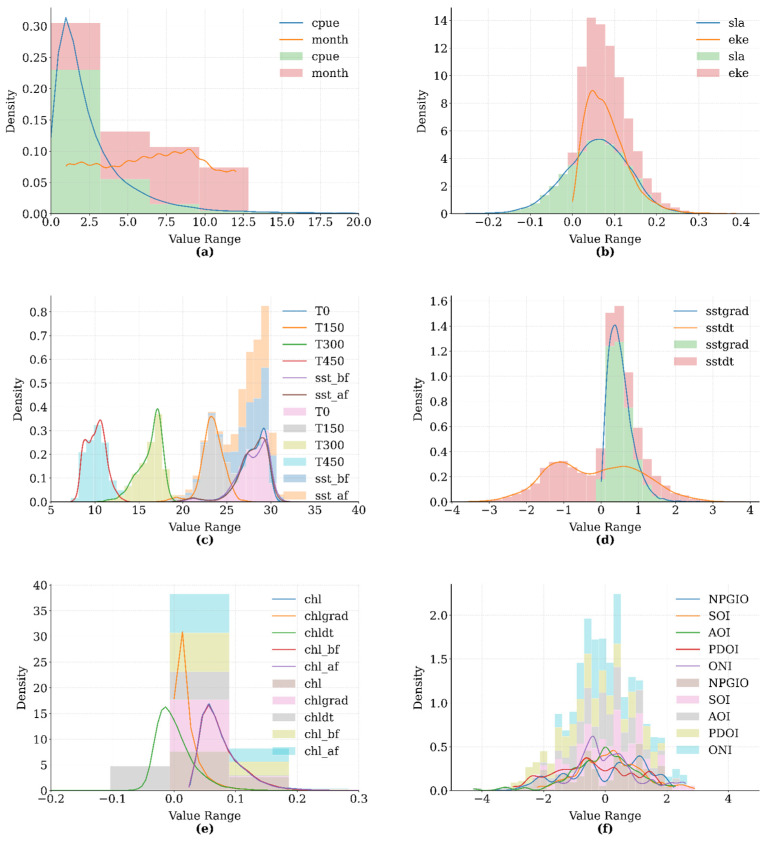
Distribution of environmental feature data: (**a**) CPUE and Month; (**b**) SLA and EKE; (**c**) Temperature profiles; (**d**) SST gradients; (**e**) Chlorophyll-a metrics; (**f**) Climate indices.

**Figure 4 biology-14-01567-f004:**
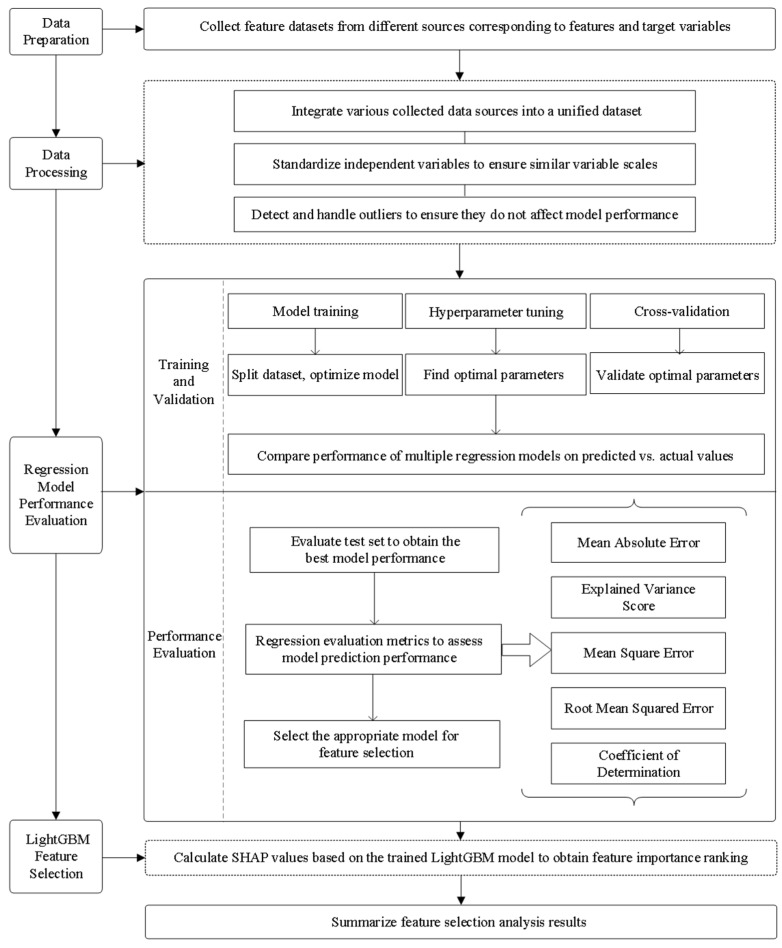
Flowchart of research methods.

**Figure 5 biology-14-01567-f005:**
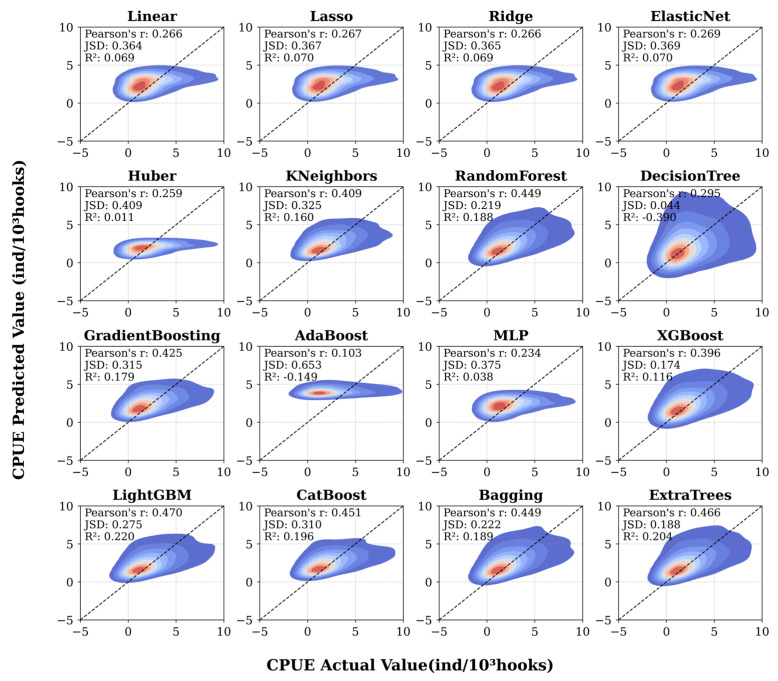
Comparison between different model predictions and actual values.

**Figure 6 biology-14-01567-f006:**
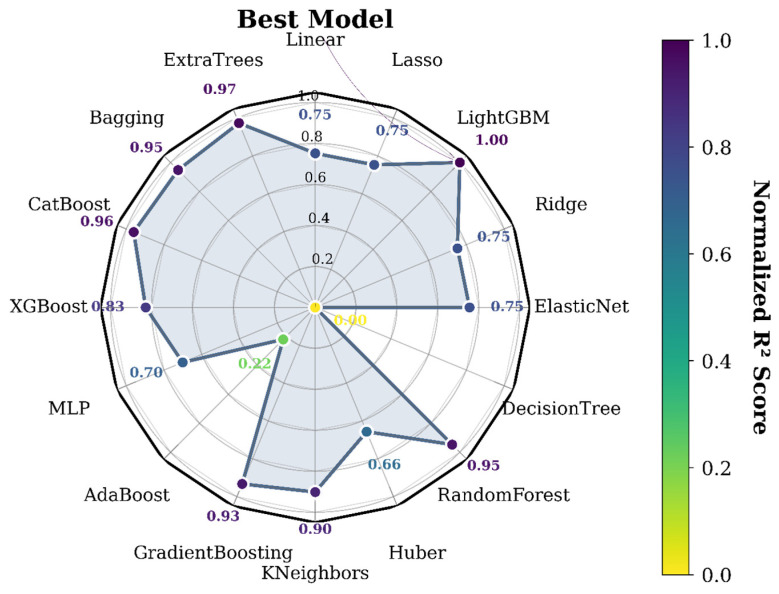
Radar charts of different model performance.

**Figure 7 biology-14-01567-f007:**
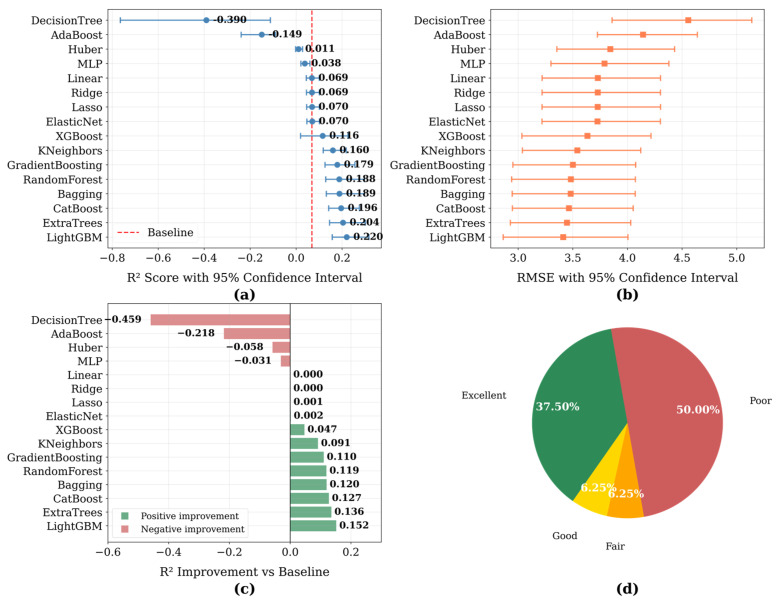
Model performance evaluation: (**a**) R^2^ scores and confidence intervals; (**b**) RMSE values and confidence intervals; (**c**) Relative R^2^ improvement; (**d**) Performance classification.

**Figure 8 biology-14-01567-f008:**
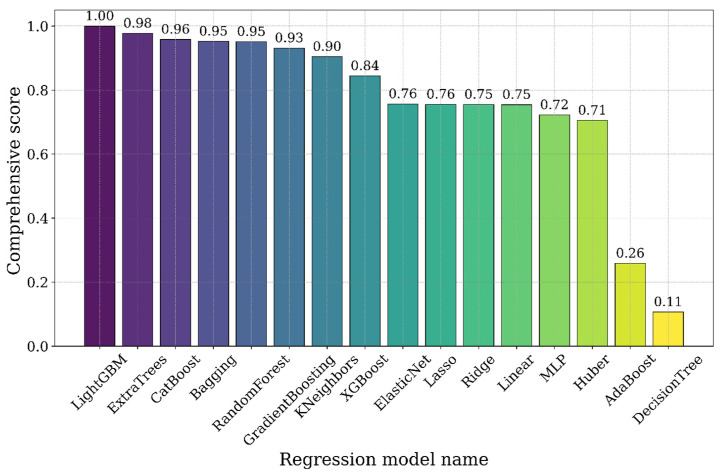
Comprehensive score of the regression model.

**Figure 9 biology-14-01567-f009:**
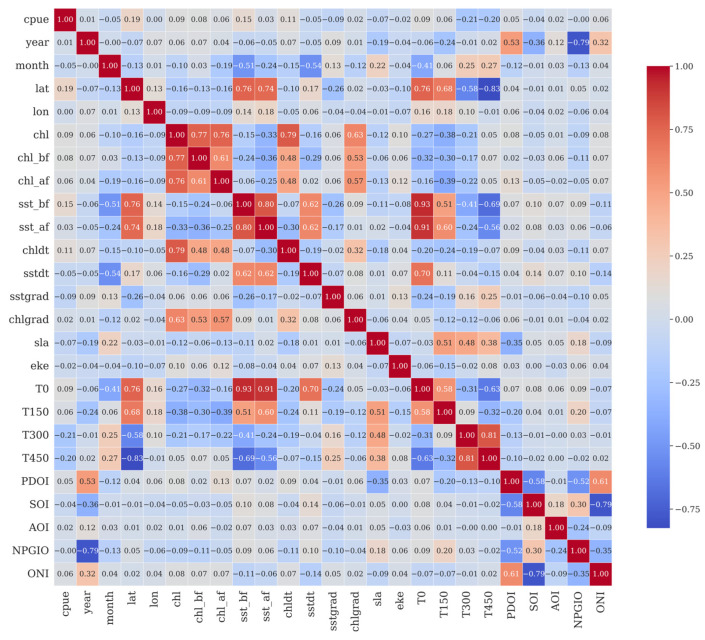
Pearson correlation coefficients between CPUE and environmental factors.

**Figure 10 biology-14-01567-f010:**
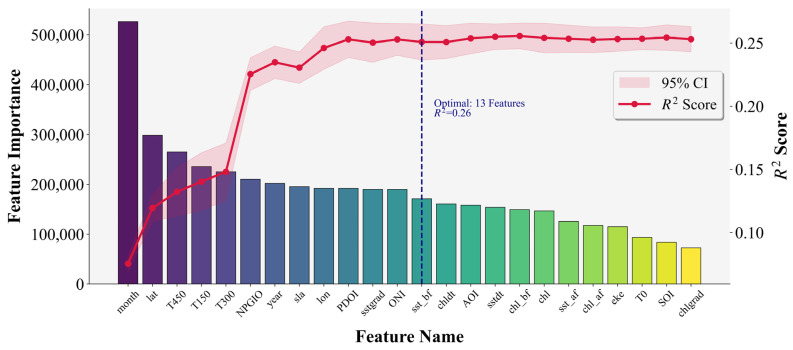
LightGBM important feature value ranking.

**Figure 11 biology-14-01567-f011:**
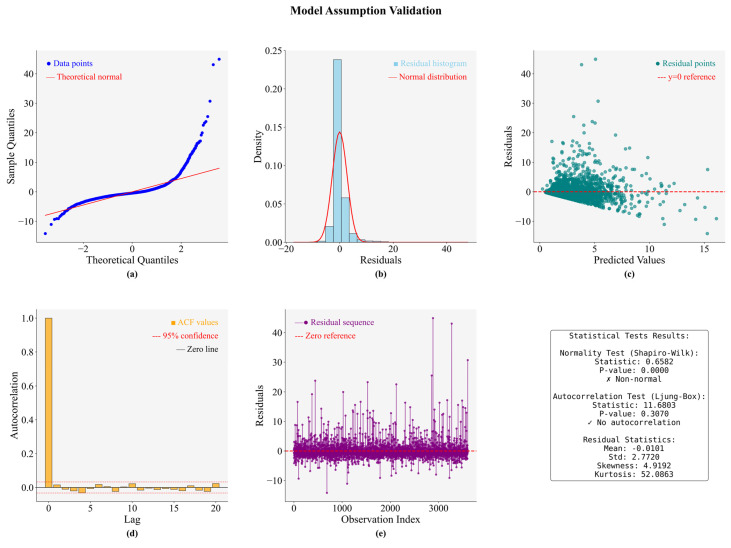
Model assumption validation results: (**a**) Q-Q plot for normality test; (**b**) Residual distribution histogram; (**c**) Residuals vs. predicted values scatter plot; (**d**) Autocorrelation function plot; (**e**) Residual sequence plot.

**Figure 12 biology-14-01567-f012:**
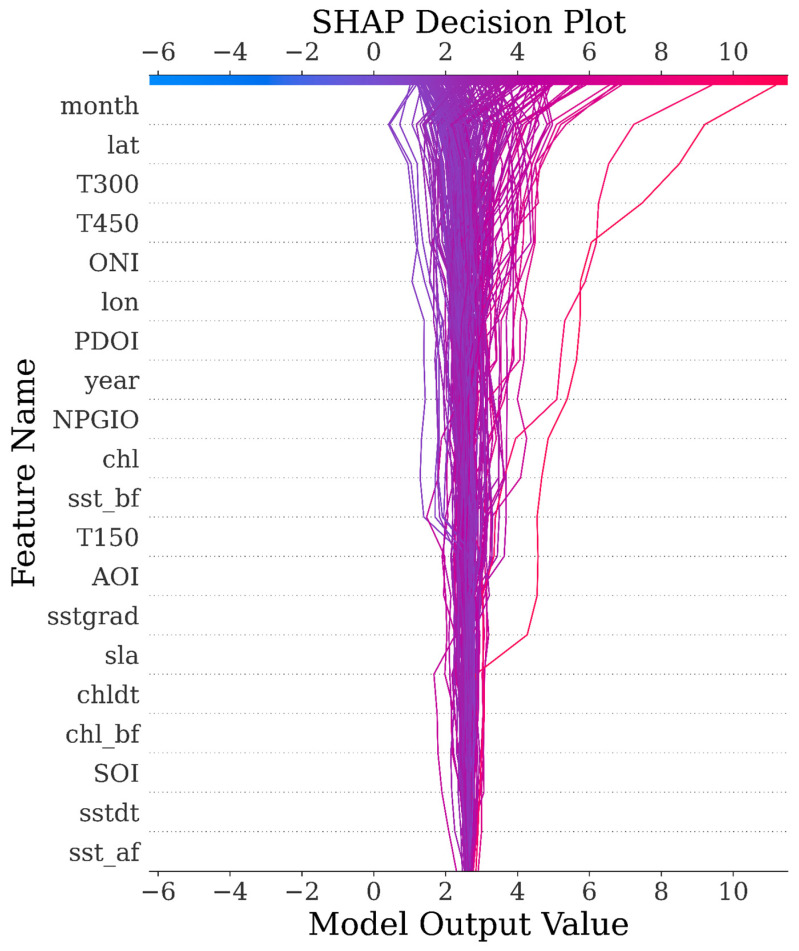
SHAP decision plot for the top 200 testing samples.

**Figure 13 biology-14-01567-f013:**
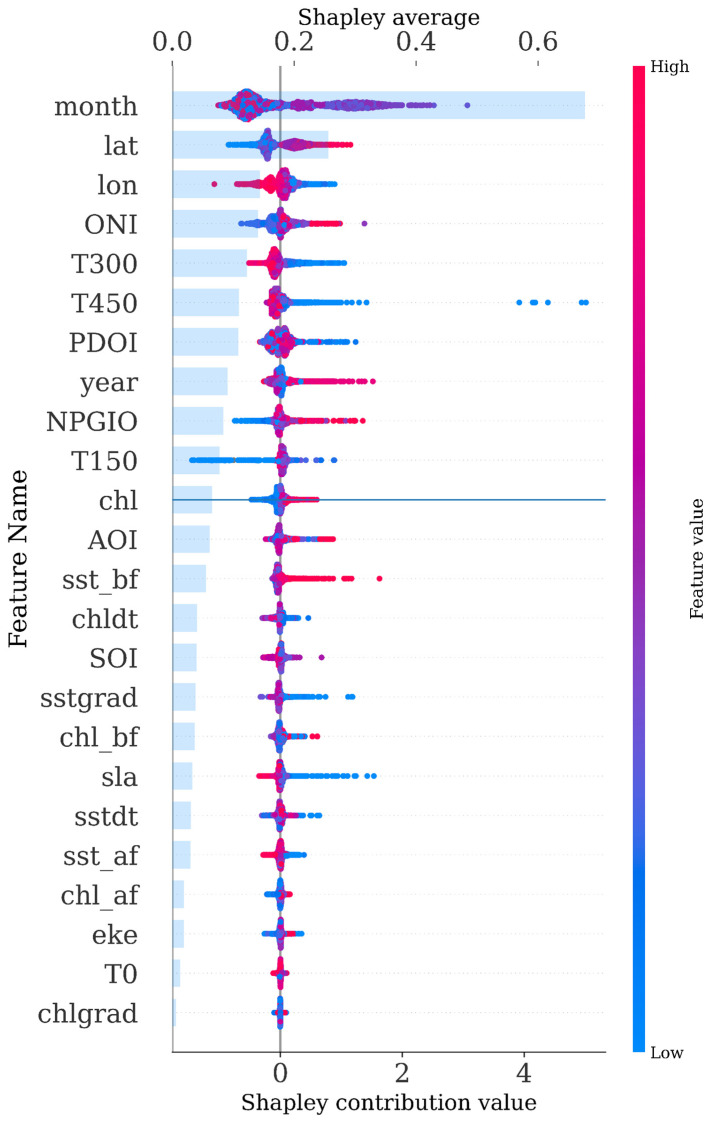
Honeycomb diagram of important features of biaxial SHAP.

**Table 1 biology-14-01567-t001:** Comparison table of MAE, MSE, RMSE, EVS, and R^2^ scores for Regression Models.

Model\Score	MAE	MSE	RMSE	EVS	R^2^
LightGBM	1.611484	11.643110	3.412200	0.221116	0.220319
ExtraTrees	1.610001	11.887523	3.447829	0.204035	0.203952
CatBoost	1.677910	12.004651	3.464773	0.197216	0.196108
RandomForest	1.634359	12.091773	3.477323	0.190333	0.190274
Bagging	1.636302	12.130093	3.482828	0.187753	0.187708
GradientBoosting	1.706000	12.256261	3.500894	0.180118	0.179259
KNeighbors	1.709640	12.541822	3.541443	0.161668	0.160137
XGBoost	1.690276	13.205482	3.633935	0.116058	0.115695
ElasticNet	1.911213	13.881904	3.725843	0.072130	0.070398
Lasso	1.911046	13.894650	3.727553	0.071288	0.069545
Ridge	1.911232	13.899188	3.728161	0.071003	0.069241
Linear	1.911564	13.906065	3.729084	0.070548	0.068780
MLP	1.872767	14.365285	3.790156	0.054812	0.038029
Huber	1.802355	14.773407	3.843619	0.057400	0.010699
AdaBoost	3.057544	18.747417	4.329829	−0.064498	−0.255421
DecisionTree	2.131442	20.764380	4.556795	−0.390487	−0.390487

**Table 2 biology-14-01567-t002:** LightGBM Optimal Parameter Table.

Parameter Category	Parameter Name	Optimal Value	Description
Basic Parameters	learning_rate	0.01	Controls the learning step size; smaller values improve model stability.
	n_estimators	800	Number of decision trees; a higher number may enhance model performance.
	max_depth	10	Maximum depth of each tree; limits complexity to reduce overfitting.
Regularization	reg_alpha	0.1	L1 regularization coefficient; encourages sparsity in the model.
	reg_lambda	10	L2 regularization coefficient; prevents excessively large weights.
Feature and Sampling	colsample_bytree	0.9	Proportion of features sampled per tree (90%).
	subsample	0.9	Proportion of samples used for training (90%), enhancing generalization.
	importance_type	gain	Feature importance evaluation method.
Tree Structure	max_bin	700	Number of bins for continuous features; higher values improve precision.
	num_leaves	35	Maximum number of leaves per tree; balances complexity and performance.
	min_child_samples	20	Minimum number of samples in a leaf to prevent overfitting.
	min_split_gain	0.2	Minimum gain required to make a split.
Randomness Control	bagging_freq	10	Performs bagging every 10 iterations.
	random_state	42	Global random seed to ensure reproducibility.
	bagging_seed	42	Random seed for the bagging process.
Computational Efficiency	n_jobs	−1	Uses all available CPU cores to accelerate training.
Evaluation Metric	Best Score	0.255	Best model performance on the validation set.

## Data Availability

No new data were created or analyzed in this study.

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
