# Peer review of "A Comparative Machine Learning Study Identifies Light Gradient Boosting Machine (LightGBM) as the Optimal Model for Unveiling the Environmental Drivers of Yellowfin Tuna (Thunnus albacares) Distribution Using SHapley Additive exPlanations (SHAP) Analysis"

_biology, 2025, doi:10.3390/biology14111567_

Round 1

Reviewer 1 Report

Comments and Suggestions for Authors

The paper presents the novel use of techniques applicable to the selection of analytical methods for T. albacares catches, a species of great importance in tropical fisheries worldwide. The methods chosen to evaluate models were well suited to the proposed objective. The graphic elements and tables presented adequately support the presentation of the results.

No major problems were detected in the writing; however, some adjustments are suggested to make the document more understandable to potential readers.

  • In the abstract, a punctuation mark at the end (line 30) should be deleted.
  • In line 338, the "x" should correspond to the formula presented in line 331, which should be lowercase, as only a reduced capital "X" was used.
  • It is useful for readers to know the total number of data points used in the analyses, particularly how much initial data (raw data) the working databases had, particularly how much CPUE data, and how much data was ultimately used in the models, that is, how much remained after cleaning.
  • It is necessary to establish the point at which the presentation of the results obtained begins because in section "3" a subtitle is presented that does not allow the reader to detect that from there the presentation of the results of the models begins.
  • I believe the first paragraph of the discussion can be deleted as it does not add anything different from what has already been stated above. It could be started from the second paragraph by adding an introductory line.

Reviewer 2 Report

Comments and Suggestions for Authors

The manuscript presents a relevant and timely study focusing on the assessment of CPUE dynamics using environmental and operational variables. However, several sections require substantial improvement in terms of methodological transparency, data description, and language clarity. The paper has potential, but major revisions are necessary before it can be considered for publication.

  1. The rationale for selecting the LightGBM model over other machine-learning or statistical methods is not discussed. The manuscript lacks validation of model assumptions (e.g., normality, independence, absence of autocorrelation). There is no explanation of whether model training and testing were separated temporally or randomly, which affects the credibility of predictive performance. Include a justification for choosing the specific modeling approach, provide diagnostics (e.g., residual analysis or cross-validation plots), and discuss model assumptions explicitly.
  2.  None of the formulas are numbered or contain units of measurement. The mathematical notation for LightGBM's objective function and regularisation terms appears simplified; the standard formulation includes first- and second-order gradient terms. Review all formulas for completeness, consistency of notation, and correctness of dimensions. Add formula numbers and define the meaning of each symbol used.
  3. The manuscript does not specify whether multicollinearity among predictors was checked before modeling. This step is crucial for the stability of the model. Model performance metrics (accuracy, RMSE, etc.) are reported without confidence intervals or comparisons to baseline models, which limits statistical interpretability. Add uncertainty quantification, include a table of model performance with confidence intervals, and compare results with a simple baseline to show improvement.
  4. The geographical scope of the research is not described in sufficient detail. The coordinates, boundaries, and temporal coverage (years, seasons, or months) should be clearly stated. A map showing the study area and fishing operation sites is essential to help readers understand the spatial extent of the data. Please specify whether the area was divided into subzones or analyzed as a single region. Provide a detailed map (with coordinates) and specify temporal coverage (years, months, or seasons).
  5. The dataset used for analysis should be described more transparently. Please specify: the number of fishing vessels and operations included in the dataset, the total number of records or observations, the spatial and temporal resolution of the data, the sources of environmental variables and whether they were measured or obtained from databases.
  6. Тhe manuscript does not describe how predictor variables were selected or screened. Please include an explanation of the feature selection process and provide a correlation matrix (e.g., Pearson or Spearman coefficients or VIF) analysis to confirm that multicollinearity among variables was controlled. Also, clarify the ecological rationale for including each environmental variable.
  7. The modeling section lacks methodological detail. The description of the LightGBM (or other algorithms used) should be expanded to include: the full list of model parameters and hyperparameters used, the procedure for hyperparameter optimization (e.g., grid search, cross-validation, or manual tuning), the metrics applied for model evaluation (e.g., RMSE, R², accuracy). Add a table of hyperparameters and describe the optimization procedure (e.g., grid search, cross-validation). Include final parameter values used in the model. Providing this information will enhance the reproducibility and transparency of the study.
  8. While the study discusses general trends, it does not provide a quantitative interpretation of how environmental factors influence CPUE. Please include variable importance ranking and, ideally, interpretability analyses (e.g., SHAP values, partial dependence plots) to demonstrate how each factor affects the predicted outcomes. Such analysis would strengthen the ecological understanding of CPUE variability.
  9. The manuscript lacks sufficient spatial analysis of fishing effort and CPUE distribution. Consider adding maps or contour plots showing how CPUE varies with longitude and latitude, as well as temporal plots illustrating seasonal or interannual variation. This will help visualize the patterns and support the conclusions drawn from the model.
  10. The Discussion section would benefit from deeper interpretation and a critical assessment of the findings. Please include: discussion of possible mechanisms behind the observed CPUE–environment relationships, identification of the main uncertainties or limitations in the dataset and modeling process (e.g., missing variables, sampling bias, or temporal coverage), suggestions for future research directions. A balanced discussion will enhance the credibility of the manuscript.
  11. A few statements in the Results and Discussion sections are assertive but not fully supported by evidence. Examples include: сlaims about “significant” influence of certain environmental variables (temperature, salinity) are made without statistical tests or confidence intervals, statements such as “the model accurately captures spatial variability” or “environmental conditions strongly affect CPUE” lack quantitative backing (no SHAP values, correlations, or sensitivity plots), the ecological interpretation of model outcomes (why specific parameters influence CPUE) is underdeveloped. Replace qualitative assertions with quantitative evidence. Include numerical values, statistical test results, or model interpretation outputs. Avoid phrases like “significant effect” unless significance is statistically demonstrated.

Comments on the Quality of English Language

1. Grammar and syntax: Inconsistent verb tenses and missing articles (“the,” “a”) occur frequently. Subject–verb agreement should be reviewed throughout.

2. Scientific style: The tone should be more formal and concise. Some expressions are conversational (“it can be seen that...”) and should be replaced with precise scientific phrasing (“The results indicate that…”).

Reviewer 3 Report

Comments and Suggestions for Authors

The manuscript under review is devoted to the influence of environmental drivers on the distribution of tuna. Tuna is an extremely important food protein resoutce for many coastal nations. Its distribution depends on multiple environmental factors, which interact in very compex ways, so the dependencies of amounts of tuna catched from these parameters are neither direct, as well as are not linear. That is why the most of models used for tuna distribution recently are not able to produce realistic prognosis of it.

Authors have selected  such parameter as catch per unit effort as the output result of their approach. They have worked with 16 different up-to-date regression models based on linear regression, decision tree models, ensemble learning techniques, and multilayer perceptron neural networks. Particularly they used the LightGBM algorithm (to address common challenges in fishery datasets, such as sample imbalance, multicollinearity among variables, and spatiotemporal dependencies). They also used SHapley Additive Explanations values to analyze the nonlinear influence mechanisms and spatial heterogeneity of environmental factors, enhancing the ecological interpretability of model outcomes. The authors selected yellowfin tuna in the western and central Pacific Ocean, with the data base of the results collected from the fishing logbooks of 43 distant-water longline vessels (China National Fisheries Corporation) for 2008-2019. The authors got the following environmental variables as inputs: Chlorophyll-a data (from NASA’s Ocean Color remote sensing platform), Sea Level Anomaly (from AVISO), Eddy Kinetic Energy, temperature–salinity profiles (from the Copernicus Marine Environment Monitoring Service), the Southern Oscillation Index, Arctic Oscillation Index (from NOAA’s Climate Prediction), the Pacific Decadal Oscillation Index (from the Climate Impacts Group at the University of Washington), the North Pacific Gyre Oscillation Index (from the Copernicus data platform) - all these sources are open and available. The authors describe in details the methods of their calculations and initial data preprocessing. The results are demonstrated by authors in the form of understandable and clear charts, schemes, diagrams. Technically the performance of this research work is really excellent. The discussion of the results is also excellent. It is deep, and it covers the main modern views on the problem.

The conclusions on “the Light Gradient Boosting Machine and Extremely Randomized Trees models achieved superior performance across all regression metrics, exhibiting excellent capabilities in nonlinear fitting and capturing complex interactions among ecological drivers, thus showing strong application potential” is proven by presented results and by exemplary and didactic discussion.

This work will be of great interest for specialists working in this field as well as for other scientists working on predictioning on the basis of multidimentional and interconnecting parameters in other, even distant fields of science. The work is excellent and the reviewer greats the authors for the fulfillment of it.

Nevertheless, there are some, though few, remarks. Authors unfortunately overuse the abbreviations. They use them even in the title, tha abstract and in the keywords. Though there is no necessity of use abbreviations in the abstract. The reviewer is to remind the authors that these parts of the paper are to be readable apart of the main text. If the subject of the paper makes the use of abbreviations necessary (as in the case of this very paper) it is better to create an abbreviations list and insert it somewhere between the keywords and introduction, or between conclusions and references. That is just a minor remark.

Reviewer 4 Report

Comments and Suggestions for Authors

This study conducted an in-depth investigation into the distribution and environmental drivers of yellowfin tuna (Thunnus albacares) in the Western Pacific Ocean using machine learning modeling and analysis. The research concept is novel, the methodology is advanced, and the data foundation is solid. The comparison of 16 regression models' performance and the introduction of SHAP methods to enhance model interpretability are particularly noteworthy. The research findings hold significant theoretical and practical value for understanding tuna fishing ground formation mechanisms and promoting precise management of fishery resources. The article is logically structured, and the figures are professional. It is recommended for acceptance after minor revisions.

Major Comments:

1.The dataset constructed for this study relies exclusively on CPUE data from Chinese distant-water longline fishing vessels. Although the data volume is substantial, the single fleet source raises the question of whether the model has learned specific operational preferences, which could affect its generalizability to different fisheries or broader maritime areas.

2.The study systematically compares the performance of 16 models, but the analysis for some underperforming models is somewhat insufficient. It is recommended to briefly discuss the potential reasons for the significant fitting deviations observed in these models, which would make the model comparison conclusions more comprehensive and profound.

3.The study effectively utilizes SHAP values to reveal the global importance of individual variables. However, the exploration of potential interaction effects between key environmental factors remains relatively shallow.

4.Please carefully check the formatting details throughout the manuscript. Ensure that citation numbers in the text correspond correctly and uniformly with the end reference list. For instance, a repeated period appears in line 32. A thorough proofreading is advised.

Round 2

Reviewer 2 Report

Comments and Suggestions for Authors

The revised version of the manuscript shows significant improvement compared to the previous version. The authors have carefully considered and incorporated all of my comments, and the overall quality of the article has improved considerably. The structure, methodology, and interpretation of the results are now clear and well-reasoned.

To further improve the readability and formatting of the manuscript, only a few minor corrections are suggested:

  1. Lines 59, 102, 133, and 205, 842: The abbreviation CPUE (Catch Per Unit Effort) is repeatedly used without an initial definition. Please provide the full form at its first occurrence.
  2. Line 112: The parameter Chl-a should be used consistently throughout the text. For example, in line 162 it appears as Chl, please unify the terminology.
  3. Figures 3, 7, and 11: Remove subfigure titles from the panels and instead include them in the main caption as (a), (b), etc.
  4. Figure 6: The lines in the Radar Chart overlap the text, please adjust layout or spacing for better legibility.
  5. Figure 9: Include the title of the legend (e.g., “Pearson coefficient”) for clarity.
  6. Figure 13: It is recommended to unify the font type and color of the text elements to maintain consistent visual style.
